# Look What I Made It Do - The ModelIT Method for Manually Modeling Nonverbal Behavior of Socially Interactive Agents

Anna Lea Reinwarth[1], Tanja Schneeberger[1], Fabrizio Nunnari[1], Patrick Gebhard[1], Uwe Altmann[2],
Janet Wessler[1]

[1]firstname_middlename.lastname@dfki.de, German Research Center for Artificial Intelligence (DFKI), Saarland
Informatics Campus, Saarbruecken, Germany

[2] firstname.lastname@medicalschool-berlin.de, Medical School Berlin (MSB), Berlin, Germany

## ABSTRACT

Nonverbal behavior of socially interactive agents (SIAs) is often automatically generated and identical across all users. This approach, though economic, might have counterproductive effects when designing applications for diverse and vulnerable populations. Also, it might negatively impact research validity and diminish the effectiveness of SIA-based interventions. This paper presents arguments for and proposes a method to model nonverbal behavior in SIAs. The ModelIT method enables researchers to ground the modelling of nonverbal behavior in psychological theories. It aims at establishing a standardized and replicable method that promotes open science practices and facilitates the creation of tailored SIAs. It is a step towards barrier-free and accessible SIA applications across diverse populations. The necessity, guidelines, and limitations of the ModelIT method are thoroughly addressed.

## CCS CONCEPTS

• **Human-centered computing → HCI design and evaluation methods**; **Interaction design process and methods**; **Accessibility design and evaluation methods**.

## KEYWORDS

Socially Interactive Agents, Nonverbal Behavior, Method, Replicability, Accessibility, Adult Attachment, Cultural Differences

**ACM Reference Format:**
Anna Lea Reinwarth[1], Tanja Schneeberger[1], Fabrizio Nunnari[1], Patrick Gebhard[1], Uwe Altmann[2], Janet Wessler[1]. 2023. Look What I Made It Do - The ModelIT Method for Manually Modeling Nonverbal Behavior of Socially Interactive Agents. In *INTERNATIONAL CONFERENCE ON MULTIMODAL INTERACTION (ICMI '23 Companion), October 9–13, 2023, Paris, France.* ACM, New York, NY, USA, 5 pages. https://doi.org/10.1145/3610661.3616549

## 1 INTRODUCTION

Nonverbal behavior plays a fundamental role in human interaction, enriching and underlining communication [10]. While humans primarily engage with other humans, in recent years these interactions include objects such as computers. In human-computer interaction,

the study of nonverbal behavior has garnered increased attention, particularly when it involves embodied and humanoid socially interactive agents (SIAs). While for specialized interpersonal tasks humans primarily engage with other humans, SIAs are becoming more and more capable of entering these spaces and performing these tasks. In a world with a rising shortage of specialists, embodied SIAs are the future of health-care [15, 27], teaching [6], coaching [35, 36], and many more previously human-dominated fields targeting diverse user groups with diverse needs. Vulnerable populations (e.g., people in in-patient facilities) make up a huge target audience for SIA applications, such as therapy-accompanying SIAs that help users identify and evaluate automatic thoughts [38]. Though modeling of nonverbal SIA behavior has been an important research focus for years [30], there is currently no standardized approach for how to model the nonverbal SIA behavior for users with diverse needs (e.g., diverse and vulnerable populations). These needs have to be identified, extracted, carefully operationalized, implemented, and validated to create SIAs that can successfully support their intended population. Therefore, this paper aims to establish and advocate for a standardized approach — the ModelIT method — for extracting nonverbal cues from literature and applying these to SIA animation. The goal is to create specialized and optimized SIAs that can confidently serve their intended purpose. It is a step towards barrier-free and equality-focused applications. The proposed ModelIT method seeks to minimize subjective biases, to ensure maximum objectivity, and to increase replicability while also considering current limitations of available technical tools.

## 2 BACKGROUND

### 2.1 Nonverbal Behavior in Human-Human-Interaction

Nonverbal behavior — including posture, movements, and facial expressions [23] — serves several important functions in human-human interaction [24]. It is a fundamental part of building rapport [39], eliciting trust [18], and establishing how a person is perceived [9]. Nonverbal behavior has been studied among the general population for a long time, but in recent years research has also focused on the many existing interpersonal differences, e.g. differences on the BIG-5 personality traits [5, 7, 11] and identified specific needs of different populations.

*2.1.1 Clinical and neurodiverse populations.* The display as well as the interpretation of nonverbal cues differ greatly when focusing on clinical or neurodiverse populations. Such populations might even perceive nonverbal cues as negative which are generally regarded and assessed as positive. For example, eye contact is usually

associated with social presence [8] and is therefore used to build trust and signify active listening to the interaction partner [12]. However, autistic people can find direct eye contact uncomfortable [32]. Consequently, nonverbal behavior should be actively adapted when interacting with an autistic person with that specific need to make them feel more comfortable.

*2.1.2 Culturally diverse populations.* Nonverbal behavior and its communicated meaning also varies significantly between different cultures [21]. When those differences during interactions with individuals from different cultural backgrounds are not considered, miscommunication is likely to occur. In some Asian countries like China, a head-shake is a nonverbal cue for "yes" and it is seen as rude to contradict another person [22]. In many central European countries, a head-shake means "no" and contradicting is a normal part of an interaction. Such differences can easily generate conflict when the interactants are not aware of them.

## 2.2 Nonverbal Behavior in Human-SIA-Interaction

Socially Interactive Agents (SIAs) are virtually or physically embodied entities capable of communicating autonomously and empathetically with human users and other agents. They can use a wide range of multi-modal behaviors [19]. Because humans tend to interact similarly with machines as with other humans, the principles of human-human-interaction can be applied to human-SIA interaction [31]. Users expect their SIA partner to act and react in certain ways and attribute human characteristics accordingly [41]. Thus, the use of certain nonverbal behavior influences how SIAs are perceived by a user [14, 33, 41]. The modeling of nonverbal SIA behavior has been an important research focus for years [30] and remains important with the technology becoming more sophisticated. The procedures of modeling nonverbal behavior of SIAs should therefore also be periodically reconsidered and standardized.

Automatic generation of nonverbal behavior has been commonly used in the field of SIAs [33]. Many SIA systems utilize pre-programmed algorithms to automatically generate gestures, facial expressions, body movements, and other nonverbal cues. While this approach guarantees a consistent user-experience, it does not fully address the many interpersonal differences in human interaction. Those differences become more relevant the more SIA applications exist for a broader spectrum of people.

*2.2.1 Clinical and neurodiverse populations.* Eye contact is generally associated with social presence [8]. Therefore SIAs are often modeled to seek eye contact with their user to build trust. When autistic people interact with a SIA designed in that way, however, eye contact could even backfire — thus compromising the application's effectiveness. It is therefore essential to model eye contact specifically for the various applications of SIAs designed for autistic people [28]. This is only one use-case in which identical automatically generated nonverbal behavior can be counterproductive.

*2.2.2 Culturally diverse populations.* Culturally varying nonverbal behavior patterns must be considered when developing SIA applications that are used interculturally. For example, a SIA-application designed for a North American user-base might completely lose its functionality when implemented in a Syrian refugee program.

## 2.3 Attachment Style

An attachment style is an internal representation and pattern of the relationship dynamics with one's close others [20, 37]. While first observed and measured in early infancy [1], the concept of adult attachment [16] argues that such learned patterns are transferred into adulthood. Various categorization systems for differentiating specific attachments styles are used by researchers. The most common distinction amongst these systems is between a secure and an insecure attachment style [1]. Some systems use continual dimensions to measure attachment style [4]: attachment anxiety and attachment avoidance. A person can score high or low on either, leading to four categories: secure (low anxiety/avoidance), preoccupied (high anxiety, low avoidance), dismissing (low anxiety, high avoidance), and fearful (high anxiety/avoidance). A person's attachment style influences their nonverbal behavior in attachment related situations [3]. Securely attached people show more nonverbal closeness (laughing, touching, gazing, and smiling) than avoidant people during an interaction with their partner [40]. People with high attachment anxiety use nonverbal cues of anger in attachment related situations [25] while people with high attachment avoidance employ distancing strategies by inhibiting nonverbal cues of their feelings [26, 34].

## 3 THE MODELIT METHOD

The ModelIT (Model it! Modeling nonverbal behavior from lITerature) method (Figure 1) is a method for standardized modeling of nonverbal behavior in SIAs. It can be applied to a wide range of use-cases and used to define SIA characteristics and specific needs of diverse user populations. It consists of five steps that lead from a research level to an applied level: (1) literature review; (2) nonverbal behavior extraction; (3) nonverbal cue operationalization; (4) nonverbal cue implementation; and (5) validation. These five steps are a necessary modeling process because nonverbal cues are frequently not defined in an applicable way in the existing literature. Especially when designing a SIA for a specific population, most nonverbal behavior is extremely complex and only vaguely described over a vast quantity of research papers. For example, people with high attachment anxiety regulate distress by seeking closeness [2]. This cannot be directly implemented into an animation but has to undergo a modeling process to be actually usable.

In our application, the nonverbal behavior of the SIA has been manually authored. However, given the possibility to express such rules with a formalism that can be programmatically interpreted, it is straightforward to foresee an automatic behavior modeling by employing a rule-based inference system [29].

## 3.1 Literature Review

The first step involves conducting an in-depth review of existing literature on nonverbal behavior, social interaction, and relevant fields of the examined use-case. The goal is to gather a comprehensive understanding of different types of nonverbal behavior examined in research and to extract them from specific papers. There are many well-documented guidelines to find papers for literature reviews and meta-analyses [17] which should be followed here for extracting the literature about nonverbal behavior. The approach should be well documented for future replicability (e.g.,

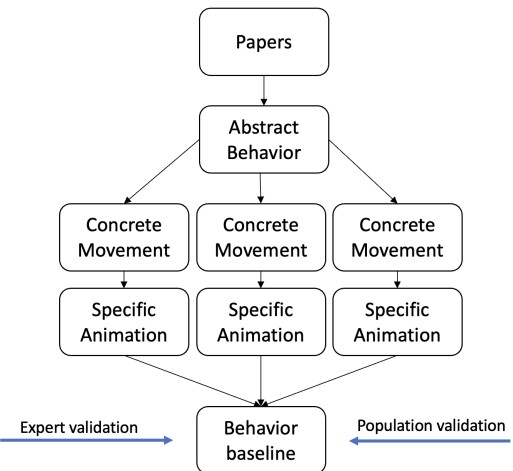

**Figure 1: ModelIT method**

used websites, search words, search date). This step provides the foundational knowledge required for all subsequent steps.

### 3.2    Nonverbal Behavior Extraction

After collecting all relevant papers, specific nonverbal behavior has to be identified and extracted. This requires taking relevant quotes about, e.g., facial expressions, body movements, gestures, and eye contact directly from papers to preserve nuanced details as accurately as possible. The quotes are then categorized and described in a spreadsheet with corresponding citations in a structured and standardized manner.

### 3.3    Nonverbal Cue Operationalization

After extracting nonverbal behavior, the next step is to operationalize it into actionable nonverbal cues for the SIA. The challenge hereby is to accurately depict the collected research findings. This includes taking the research quotes, transforming abstract behavior into concrete movements and adding them to the spreadsheet.

### 3.4    Nonverbal Cue Implementation

The list of operationalized nonverbal cues can now be implemented for the specific used animation system. The cues should be examined one by one and individually translated into available animations. This can be challenging because many existing SIA animation systems only have predefined fixed animations. Those need to be individually evaluated whether they actually convey the intended meaning. For example, an animation called "puzzled" is not necessarily categorized by users as puzzled. The end result should be a spreadsheet with directly usable animations and rules as how to use them. The actual animation process can now easily be done while following the predefined rules.

### 3.5    Validation

The last step consists of validating the resulting SIA behavior. This step is crucial to actually test the intended effect of customized SIAs. Two separate types of validation should be considered: validation

of the accuracy of the nonverbal behavior (experts) and validation of the functionality of the nonverbal behavior (target population). If resources are limited, validating if the nonverbal behavior fulfills its function for the specific target population should be prioritized. Accuracy can then be validated informally and consensus-based by at least two experts.

## 4    APPLICATION OF THE MODELIT METHOD

We applied the ModelIT method to the modeling of nonverbal behavior of SIAs representing people with different attachment styles. The following section does not show the documentation process exhaustively, but it exemplifies the ModelIT method step by step and highlights its importance.

### 4.1    Literature Review

The goal of the literature review was finding nonverbal behavior of different attachment styles. The first step was implementing a search strategy. Search terms were for example: "attachment style nonverbal", "attachment presentation nonverbal", "attachment style behavior", "adult attachment behavior",… The used sites were "Web of Science" and "GoogleScholar". We decided to only use nonverbal behavior of attachment style in adults and not children. The chosen categorization system for the SIA was: secure, dismissing, and preoccupied. Importantly, we still wanted to utilize literature referring to every other attachment style classification system. This strategy was adopted to extract as much information as possible because literature on this specific topic is sparse. The classification systems are comparable and therefore convertible into each other, which was done in a later step. However, it is crucial to note that in different scenarios such comparability might not be given, thus necessitating a comprehensive documentation of inclusion/exclusion criteria.

### 4.2    Nonverbal Behavior Extraction

The result of our literature review was a list of papers with information about nonverbal behavior for the dimensions attachment anxiety (high) and attachment avoidance (high/low) as well as the categories secure, insecure generally, preoccupied, and dismissing attachment. We organized them into a spreadsheet and sorted the nonverbal behavior into this framework with direct quotes.

### 4.3    Nonverbal Cue Operationalization

The step from abstract nonverbal behavior to concrete movements is very complex due to the nature of the preexisting literature. Nonverbal behavior is often ambivalent or not precisely defined and cannot be categorized into replicable movements — it needs to be further operationalized and formalized. For example, dismissing attachment compared to secure is associated with less movement in general and less movement complexity [3]. Applying such findings to modeling SIA behavior shows the complexity. We defined a baseline behavior that this behavior varies from. Here, the baseline is the nonverbal behavior of secure attachment. Then we decided which movements will be shown less, for example, full body, hand, head, or even every kind of movement. The concept of movement complexity is also ambivalent and could apply to many facets and types of movements, making further formalization necessary. We

Table 1: Example: ModelIT spreadsheet for high Attachment Avoidance

| Reference | Extraction | Operationalization | Application |
|---|---|---|---|
| Mikulincer & Shaver (2005) [26] | "blunted affect" | smiles without teeth | emot.smile (not emot.happy) |
| | | neutral facial expression as baseline | emot.bored as baseline |
| | | low frequency of facial expression changes | emot.bored > 80 percent of the time |
| | | only small movements | sad03 (sigh, tiny head-shake, no full-body movement) |
| Fraley & Shaver (1998) [13] | "frequent avoidant behaviors" | turning away | lookto.right80.01 |
| | | looking away from the user | lookat.07.01 |
| | | keeping distance hand movement | number.handl.5 |
| | | avoiding eye contact | eye contact < 50 percent of the time |

chose standard movements for secure attachment style and then searched for similar movements with less intensity (e.g., "shrugging motion" as a shoulder movement with simultaneous arm movements vs. shoulder movement only). Quantifying "intensity" while working with fixed animations can be challenging, and should rely on the consensus of several raters. Individuals with insecure attachment tend to show discrepancies between verbal content and nonverbal behavior [3]. Therefore, the content of the spoken text needs to contradict the content of the nonverbal behavior while considering intercultural differences: In a central European culture, a discrepancy is created when the SIA says "yes" while shaking their head. This effect vanishes for a Chinese SIA [22]. Further, for our example, individuals typically only express nonverbal behavior patterns of attachment style in specific attachment related situations, e.g. while taking about their early caregivers [3]. This is addressed by defining attachment related situations based on research and labeling them in the action/interaction. The SIA then displays the modeled behavior only during those labeled sections.

### 4.4 Nonverbal Cue Implementation

Every operationalized nonverbal cue on the spreadsheet was matched with available animations in Vuppetmaster[1] – a tool for modeling SIA behavior. This resulted in a detailed and comprehensible spreadsheet used to animate videos of SIAs displaying nonverbal behavior of secure, dismissing, or preoccupied attachment. Table 1 shows an excerpt from our spreadsheet, including the referenced paper, abstract concept, operationalization, and application in the form of a Vuppetmaster command. See Figure 2 for examples.

### 4.5 Validation

It must be validated that e.g., dismissingly attached people actually show the modeled nonverbal behavior. This should involve experts like psychotherapists. Then, the functionality of the SIAs must be validated and rated by people with the corresponding attachment.

## 5 DISCUSSION

In this paper, the five-step ModelIT method is proposed for modeling nonverbal behavior in SIAs: (1) literature review; (2) nonverbal behavior extraction; (3) nonverbal cue operationalization; (4) nonverbal cue implementation; and (5) validation. Scientific literature frequently lacks information about nonverbal behavior of specific populations. When available, it can be ambivalent and vague. Therefore, it has to be carefully examined and evaluated to operationalize it into actionable cues. This should be done transparently while documenting each modeling step and whether it is research- or consensus-based. The ModelIT method gives researchers a strong

[1]https://vuppetmaster.de/

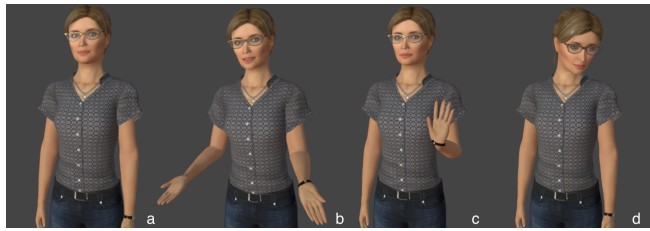

Figure 2: Examples of nonverbal Cues for a: idle behavior, b: secure attachment, c: preoccupied attachment, d: dismissing attachment

theoretical foundation and transparent guide to ground their modeling of nonverbal behavior in psychological theories.

### 5.1 Limitations and Future Work

While the ModelIT method is based on a thorough theoretical foundation, one goal for future work is to empirically compare it to automatic nonverbal behavior generation. Our approach to modeling nonverbal behavior is more time-consuming than automatic generation. But the output of the ModelIT method can be used to employ a rule-based inference system for automatic behavior modeling [29]. Additionally, the benefit for vulnerable and diverse populations justifies additional costs and time investment. Despite best efforts to produce a standardized procedure, the proposed method can still (though less) be impacted by human biases. Especially when the team of researchers modeling the SIA is homogeneous (e.g., concerning cultural background, gender identity). This must be counteracted with the proposed documentation to make human error at least traceable and therefore hopefully solvable.

## 6 CONCLUSION

There is a growing need for careful modeling of nonverbal SIA behavior to further a barrier-free and intercultural world. Interpersonal differences need to be considered in order to minimize biases. Moreover, transparent documentation is an essential step towards open science and accessible SIA applications. Therefore, this paper introduced the ModelIT method for modeling appropriate nonverbal behavior in SIAs, enhancing their ability to engage with specific user populations that have been overlooked in the past.

## ACKNOWLEDGMENTS

This work is funded by the German Federal Ministry for Education and Research (BMBF) within the UBIDENZ project (funding code 13GW0568D). Thanks to Shailesh Mishra for making the Taylor Swift joke that turned into the title of this paper.

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
