# OpenReview forum: "Look What I Made It Do - The ModelIT Method for Manually Modeling Nonverbal Behavior of Socially Interactive Agents"
_ACM.org/ICMI/2023/Workshop/GENEA — GENEA Workshop 2023_

### Official Review · Reviewer_no7e · 2023-08-08
**Methodology for researchers of gesture generation field for good practice.**

**Rating:** 6
**Confidence:** 5

**Review:**

The paper aims to create specialized and optimized SIAs for diverse needs and specific populations who might be vulnerable and need carefully implemented SIAs. The authors propose ModelIT, a standardized method of nonverbal behavior modeling based on psychological theories. Their method consists of five steps focusing on documentation for reproductivity: literature review, manual extraction of nonverbal behavior, operationalization of nonverbal cues, implementation, and validation (based on accuracy and function) nonverbal cues. Their method is explained for the application of modeling of different attachment styles.

Strengths:
- The motivation of the study is clear and a thorough background study is done.
- The authors provide a theoretical and transparent guideline for manual modeling of nonverbal behavior.

Weaknesses:
- Though the clear guideline may help researchers perform manual modeling in a more reproducible way, the novelty of the work is questionable. The proposed method seems to be an already known method that serves more as a reminder to the researcher of the field of gesture generation.
- The authors state that their method can enhance SIAs’ ability to engage with specific user populations (with the basis that it is better than automatic generation models using recent ML/DL techniques). However, no evidence was provided to support their statement (no previous work cited and no evaluation was performed). Thus, this statement remains as an assumption awaiting to be tested.

Recommendations and Questions:
- Please cite some previous works concerning “SIA systems based on pre-programmed algorithms”. Recent works have started to model the interpersonal dynamics within the interaction.
- Are there any previous works demonstrating the problem of culturally varying or interchanging nonverbal behavior patterns?
- For the validation, the authors state that the animated gestures must be validated along the aspects of accuracy (by expert) and function (by concerned population). It would be best to address both of them but it is an ideal case. If only one of the two could be maximized, which one should researchers aim to resolve?
- Maybe a typo in line 7 of section 2.2.1 “... eye gaze essential, especially …”, please verify.
- Typo in line 5 of section 3.1. “… nonverbal behavior from.”
- Please verify the uniformity of the English. Some words are in British English (e.g.“modelling”, “concretised”) and others are in American English (“Behavior”, “modeling”, “operationalize”).

---

### Official Review · Reviewer_4dhj · 2023-08-08
**A good paper, highlighting issues that are often not considered, also providing a framework to help future work.**

**Rating:** 7
**Confidence:** 3

**Review:**

### Summary
This work introduces a framework for rule-based interactive agent gesture generation where the target audience for the interactive agent is an important focus to ensure the interactive agent is appropriate for the audience in which it is deployed. The importance of this framework is introduced with examples of diverse and vulnerable groups where the same interaction and gesture style across all individuals could  diminish the effectiveness of the interaction. A five-step process is introduced to ensure interactive agents are specifically designed to fulfil diverse needs. (1) literature review; (2) nonverbal behaviour extraction; (3) nonverbal cue operationalization; (4) nonverbal cue implementation; and (5) validation. A workflow example of this process is provided and discussed in the work.

### Strengths
This paper highlights an important problem, often overlooked when generating gestures for interactive agents. I think this does a good job of stating a number of use cases for which the method can be used. Although the paper explicitly suggests a spreadsheet and linking to Vuppetmaster, I do not see any issue with this framework being extended to other gesture-generation languages/systems in future. The paper is well-structured and easy to read.

### Weaknesses
1) While I understand the evaluation of this method is difficult, some examples of the generated gestures for a few use cases would be useful to include and compare, particularly video if possible.
2) I have a small concern regarding contradictory instructions that may occur. For example in Table 1, “lookto.right80.01” and “lookat.07.01”, it is unclear what behaviour may occur.
3) Table 1 is not discussed or referenced in the text.
4) I feel more discussion on the choice of actions/implementation from the operationalization stage could be useful.

---

### Decision · Program_Chairs · 2023-08-11

**Decision:**

Accept

**Comment:**

Both reviewers agree on accepting this paper stating that the authors address an important problem for agent gesture generation. The chairs agree with their decision to accept the paper for the GENEA workshop. Please read the reviews and update the paper for the camera-ready version. The evaluation of the proposed method could be challenging but it will better demonstrate the advantage of the method by showing some examples of the generated gestures of the method and comparing them to the gestures created by other techniques (automatic ML/DL models), particularly providing a video (if possible) will help.